# L-Citrulline Supplementation Restrains Ferritinophagy-Mediated Ferroptosis to Alleviate Iron Overload-Induced Thymus Oxidative Damage and Immune Dysfunction

**DOI:** 10.3390/nu14214549

**Published:** 2022-10-28

**Authors:** Tongtong Ba, Dai Zhao, Yiqin Chen, Cuiping Zeng, Cheng Zhang, Sai Niu, Hanchuan Dai

**Affiliations:** College of Veterinary Medicine, Huazhong Agricultural University, No.1 Shizishan Street, Wuhan 430070, China

**Keywords:** L-citrulline, thymus, iron overload, ferritinophagy, ferroptosis

## Abstract

L-citrulline (L-cit) is a key intermediate in the urea cycle and is known to possess antioxidant and anti-inflammation characteristics. However, the role of L-cit in ameliorating oxidative damage and immune dysfunction against iron overload in the thymus remains unclear. This study explored the underlying mechanism of the antioxidant and anti-inflammation qualities of L-cit on iron overload induced in the thymus. We reported that L-cit administration could robustly alleviate thymus histological damage and reduce iron deposition, as evidenced by the elevation of the CD8^+^ T lymphocyte number and antioxidative capacity. Moreover, the NF-κB pathway, NCOA4-mediated ferritinophagy, and ferroptosis were attenuated. We further demonstrated that L-cit supplementation significantly elevated the mTEC1 cells’ viability and reversed LDH activity, iron levels, and lipid peroxidation caused by FAC. Importantly, NCOA4 knockdown could reduce the intracellular cytoplasmic ROS, which probably relied on the Nfr2 activation. The results subsequently indicated that NCOA4-mediated ferritinophagy was required for ferroptosis by showing that NCOA4 knockdown reduced ferroptosis and lipid ROS, accompanied with mitochondrial membrane potential elevation. Intriguingly, L-cit treatment significantly inhibited the NF-κB pathway, which might depend on restraining ferritinophagy-mediated ferroptosis. Overall, this study indicated that L-cit might target ferritinophagy-mediated ferroptosis to exert antioxidant and anti-inflammation capacities, which could be a therapeutic strategy against iron overload-induced thymus oxidative damage and immune dysfunction.

## 1. Introduction

Iron is an essential micronutrient and critical for cell metabolism, proliferation, differentiation, and survival [1,2]. Iron is involved in fundamental metabolic processes such as cellular respiration, DNA synthesis and DNA repair, oxygen transportation, citric acid cycle, and immunity [3,4,5]. Systemic iron homeostasis is crucial to terrestrial life. However, iron overload (IO) leads to disorder of the iron metabolism. Patients with conditions such as aplastic anemia, myelodysplastic syndrome, sickle-cell disease, as well as transfusion-dependent patients with thalassemia major, have the characteristics of IO, and excessive iron can be cytotoxic, resulting in the inducing of global oxidative damage and lipid peroxidation by generating reactive oxygen species (ROS) [6,7,8]. Moreover, iron overload eventually leads to serious chronic consequences including hemorrhagic inflammation, hepatic cirrhosis, diabetes, innate immune dysfunction, as well as growth arrest and cell death [1,9,10,11,12]. Iron overload-catalyzed oxidative stress and immune damage have attracted much interest because of the sophisticated mechanism leading to cell death and its association with diseases [13].

The thymus is considered as the central lymphoid organ for production of diverse immune cells and provides proper environments for T lymphocytes maturation, differentiation, and function [14]. The mature T cells in the thymus can migrate to the lymph nodes, aiding the immune system to fight the disease [15,16]. The thymus is extremely sensitive to pressure. T-2 toxin, sulfoxaflor, Mycoplasma gallisepticum, and high concentrations of monosodium glutamate can induce thymus oxidative damage and immune dysfunction [17,18,19,20]. Iron is required for the proliferation and function of the thymus. T lymphocytes can synthesize ferritin, which acts as a “mobile” and easily “mobile” iron-storage compartment [21]. However, the ability of the ferritin of lymphocytes to chelate excess iron is poor, which might result in immune function disorder [22]. The presence of transferrin receptor on immature, proliferating thymocytes highlights the importance of iron to T cell development [23]. Transferrin can be exploited to modify immune responses and provide a profound cytoprotection against proapoptotic and cytotoxic signals in the thymus [24,25]. Iron deposition in most organs has been described. Iron deficiency in the thymus is well established. Iron deficiency contributes to a reduction in peripheral T cells and atrophy of the thymus [26]. However, the relationship between iron overload and the damaging of the thymus is rarely described. The mechanism of thymic immune damage induced by iron overload is unclear.

Ferritinophagy, a selective form of autophagy, contributes to the initiation of ferroptosis through degradation of ferritin, which triggers labile iron overload (IO), lipid peroxidation, membrane damage, and cell death [27]. Mounting studies show that nuclear receptor coactivator 4 (NCOA4) is an autophagosome component that participates in the process of ferritinophagy [28,29,30] and has gradually been considered as a key molecule promoting ferroptosis. NCOA4 depletion can inhibit ferroptosis by eliminating the accumulation of intracellular free iron [30,31]. Ferroptosis is characterized by high iron-dependent lipid peroxidation, and reactive free iron has been identified as the final executor of ferroptosis [32,33,34]. Ferroptosis causes cells to emit damage-associated molecular patterns (DAMPs) and alarmins, which increases cell death and triggers a cascade of inflammatory reactions [35,36,37,38]. Ferritinophagy-mediated ferroptosis plays an essential role in the balance of iron metabolism, contributing to preventing tissue injury and immunological responses. Studies indicated that targeting ferritinophagy-mediated ferroptosis in cardiomyocytes may be a therapeutic strategy for preventing sepsis-induced cardiac injury [39]. Ischemic stroke poses a significant health risk due to its high rate of disability and mortality. NCOA4 deletion notably abrogated ferritinophagy caused by ischemia/reperfusion neuronal injury and thus inhibited ferroptosis [40]. NCOA4 mutation reduced ferroptosis and weakened anticancer immune responses in clear cell renal carcinoma [30,41]. Additionally, ferritinophagy-mediated ferroptosis is involved in liver fibrosis [42], acute kidney injury [43], and cancers [44,45].

As a nonessential amino acid, L-cit functions as an intermediate in the urea cycle and the citrulline-nitric oxide cycle as well. L-cit may serve as a precursor for the endogenous synthesis of argininosuccinate, which can augment the synthesis of NO [46,47]. L-cit possesses antioxidation, anti-inflammatory, and immunity elevation properties. It can efficiently protect DNA from oxidative damage, supplement essential nutrients, and boost immunity [48]. It is shown that endogenous citrulline seems to play a key role in regulating endothelial and immune functions by regulating nitric oxide production [49]. L-cit therapy enhanced the number of Tregs cells of infant rats by promoting both IL-10 and TGF-β1 production [50]. Citrulline was identified as an innate immune signaling metabolite that engages a metabolic checkpoint for proinflammatory responses [47] and has been proven to exert anti-inflammation and antioxidant qualities. However, the biological function of citrulline remains obscure in the thymus against iron overload.

In the current study, the antioxidant capacity and immune modulatory of L-cit on the mouse thymus challenged with iron overload were investigated. Our data provided evidence that L-cit alleviated thymic oxidative damage and immune dysfunction by enhancing Nrf2 (nuclear factor erythroid 2-related factor 2) nuclear translocation and suppressing the NF-κB signaling pathway, which might rely on inhibiting ferritinophagy-mediated ferroptosis against iron overload.

## 2. Materials and Methods

### 2.1. Animal Treatment

All animal experiment procedures were approved according to the guidelines of the Animal Ethical Committee of Huazhong Agricultural University (License No. HZAUMO-2020-0079) and complied with the ARRIVE guidelines and the National Research Council’s Guide for the Care and Use of Laboratory Animals. Male C57BL/6J mice (aged 4 weeks, weighing 19.52 ± 0.86 g) were purchased from Huazhong Agricultural University Experimental Animal Center with the license number SYXK (Hubei) 2020-0084. C57BL/6 mice were housed in a temperature and humidity-controlled room (22 °C, 50% relative humidity) with 12 h light/dark cycles and randomly divided into six groups (*n* = 8), namely, control group, iron overload group (IO), L-cit group (L-cit _1 g/kg_), IO + L-cit _0.5 g/kg_ group, IO + L-cit _1 g/kg_ group, and IO + L-cit _2 g/kg_ group. The mice were intraperitoneally injected with iron dextran 50 mg/kg (Aladdin, China) [51,52]. L-cit and an equal volume of saline were given by gavage administration every day. Doses of L-cit were set to 0.5 g/kg·bw, 1 g/kg·bw, and 2 mg/kg·bw. After two weeks, the mice were anesthetized with an intraperitoneal injection of sodium pentobarbital (30 mg/kg) and sacrificed. The serum and thymus were collected for future analysis.

### 2.2. Hematoxylin and Eosin Staining (H&E Staining)

Thymus was fixed in formalin, then dehydrated with ethanol and embedded in paraffin wax. The paraffin wax thymic tissue blocks were cut into 5 μm thick and stained with hematoxylin and eosin. Stained tissue sections were thoroughly examined by light microscopy (Olympus BX51, Tokyo, Japan) with an attached digital camera (Olympus DP72, Tokyo, Japan).

### 2.3. Prussian Blue Staining

Prussian blue staining was used to examine the iron deposits. The thymus tissue section was first deparaffinized and hydrated, and then incubated for 3 min in a working solution made up of equal parts potassium ferrocyanide solution and hydrochloric acid solution. Finally, the iron deposits in the cytoplasm were seen using an optical microscope (Olympus DP72, Tokyo, Japan).

### 2.4. Cell Viability Assay

mTEC1 cell viability was assessed by CCK-8 assay (Yeasen, Shanghai, China). mTEC1 cells (5 × 10^5^ cells/mL) were suspended in a 96-well plate with enriched RPMI-1640 medium containing 1% penicillin/streptomycin and 10% fetal bovine serum (Gibco, Grand Island, NY, USA). The cells were maintained at 37 °C in a humidified atmosphere with 5% CO_2_ and stimulated with/without FAC (ferric ammonium citrate) (Sigma, St. Louis, MI, USA) or L-cit (Solarbio, Beijing, China). After 24 h or 48 stimulation, 10 μL CCK-8 solution was added to each well and then incubated for 60 min at 37 °C with 5% CO_2_. A microplate reader (Thermo Fisher, Waltham, MA, USA) was used to measure the absorbance at 450 nm.

### 2.5. siRNA Transfection

mTEC1 cells were seeded in 6-well plates and cultured in 1640 medium containing 1% penicillin/streptomycin and 10% fetal bovine serum (Gibco). The cells were maintained at 37 °C in a humidified atmosphere with 5% CO_2_ and stimulated with 200 μM FAC or 2 mM L-cit (Solarbio, Beijing, China) for 24 h. X-tremeGENE Transfection Reagent (Roche, Basel, Swiss) was used to transfect siNCOA4 (GenePharma, Suzhou, China) according to the manufacturer’s recommendations. The sequence of the siRNA used is as follows: siRNA-NCOA4 5′-CCAUCAGGACACAUGUAAATT-3′, 5′-UUUACAUGUGUCCUGAUGGTT-3′; siRNA-NC 5′-UUCUCCGA

ACGUGUCACGUTT-3′. 5′-ACGUGACACGUUCGGAGAATT-3′. The cell pellets and culture supernatants were collected at the indicated time for further studies.

### 2.6. Determination of Iron Concentration

mTEC1 cells (1 × 10^7^) were harvested after incubation with/without FAC and L-Cit for 24 h. The cells were washed with cold PBS and then lysed with lysis buffer on a shaker for 2 h. The samples were centrifuged at 12,000 r/min for 10 min to remove insoluble materials. An amount of 4.5% potassium permanganate solution was mixed with the buffer of the Iron Assay Kit (Applygen, Beijing, China) at a ratio of 1:1. The reagents were added in the control tube, standard tube, and sample tube. The samples were incubated at 60 °C for 1 h, then cooled and added to iron ion detection reagents. The samples were incubated at room temperature for 30 min. Absorbance was measured at 550 nm by a microplate reader (ThermoFisher, Waltham, MA, USA).

### 2.7. Determination of MDA Content, SOD, GSH-Px, and LDH Activity

The thymus tissue and cell homogenate were prepared using an ultrasonic disintegrator. The samples centrifuged, and the supernatants were collected. The malondialdehyde (MDA) content, the activity of superoxide dismutase (SOD), glutathione peroxidase (GSH-Px), and lactate dehydrogenase (LDH) in the serum and supernatants were explored using the commercially available kit (Jiancheng, Nanjing, China). The absorbance of optical density at 532 nm (MDA), 450 nm (SOD, LDH), and 412 nm (GSH-Px) was measured using a microplate spectrophotometer (ThermoFisher, Waltham, MA, USA). All procedures were performed according to the manufacturer’s instructions.

### 2.8. Real-Time Fluorescence Quantitative PCR

Total RNAs were extracted with Trizol (Invitrogen, Carlsbad, CA, USA) according to the manufacturer’s instructions, and RNA concentrations were determined using Nanodrop 2000 (ThermoFisher, Waltham, MA, USA) with 260/280 ratios of 1.8–2.0 considered acceptable. RNAs were reverse-transcribed with M-MLV reverse transcriptase (Invitrogen, Carlsbad, CA, USA). The qPCR assay was performed with the SYBR Green PCR kit (Toyobo, Osaka, Japan) in a LightCycler^®^ 96 Real-time PCR instrument (Roche, Basel, Swiss). Glyceraldehyde-3-phosphate dehydrogenase (GAPDH) was used as an internal standard. The quantitative date was analyzed using the 2^(−ΔΔCt)^ method, and each sample was conducted at least triplicate. The primers for NCOA4 gene were prepared by the company (Tsingke, Wuhan, China), and the primer sequence is: NCOA4 5′-CACAATGAGCCTAAGCCAGC-3′. 5′-CCCCTCTGTAGCAACCATCC-3′. GADPH 5′-AAATGGTGAAGGTCGGTGTGAAC-3′ and 5′-CAACAATCTCCACTTTGCCACTG-3′.

### 2.9. Western Blot Assay

Proteins were extracted from thymus tissues using RIPA lysis buffer (Beyotime, Beijing, China) containing 1 mM PMSF (Beyotime, Beijing, China) and 1 mM phosphatase inhibitor (Beyotime, China) and quantified using the BCA protein kit (Beyotime, Beijing, China). The total proteins were added to a 12% sodium dodecyl sulfate–polyacrylamide gel electrophoresis (SDS-PAGE) gel for separation and then transferred to polyvinylidene fluoride (PVDF) membranes. After blocking with 5% skimmed milk powder for 2 h, the membranes were incubated with the primary antibody overnight at 4 °C. The primary antibodies were removed by washing three times. The secondary antibodies were incubated at room temperature for 90 min and then washed three times with TBST. Electrochemiluminescence (ECL) imaging system (Biotanon-5200, Shanghai, China) was used to scan the immunoreactivity bands of the target proteins, and Image J software (National Institutes of Health, Bethesda, MD, USA) was used to quantify the digital data of the integrated optical density. Primary antibodies including TfR1 (1:1000), p65 (1:500), LC3 I/II (1:1000), and GAPDH (1:5000) were purchased from Proteintech (Wuhan, China). NCOA4 (1:2000), FTH (1:2000), and GPX4 (1:2000) were purchased from Abclonal (Wuhan, China). IL-6 (1:1000), IL-1β (1:500), and TNF-α (1:1000) were purchased from Wanleibio (Shenyang, China), and p-p65 (1:1000) and Nrf2 (1:1000) were purchased from Santa Cruz.

### 2.10. Mitochondrial Membrane Potential (MMP) Measurement

5′,5′,6’,6′-tetrachloro-1′,1′,3,3′-tetraethyl-imidacarbocyanine iodide (JC-1) probes were used to measure the MMP using a commercially available kit (Yeasen, China). JC-1 forms a polymer in healthy mitochondria and displays red fluorescence. It will exist in monomer form and emit green fluorescence when the mitochondrial membrane potential lowers. The procedure was performed according to the manufacturer’s instructions.

### 2.11. Cytoplasmic ROS and Lipid ROS Measurement

The level of intracellular ROS was evaluated by the uptake of 2,7-Dichlorodihydrofluorescein diacetate (DCFH-DA). DCFH-DA is a broad biomarker of oxidative stress. The mTEC1 cells were seeded in 8-well plates with 1.5 × 10^4^ cells/well and treated with the designated conditions. After the indicated time of treatment, cells were double stained with 50 μM DCFH-DA. DCFH-DA was subsequently oxidized to create 2′,7′-dichlorofluorescein (DCF), a bright fluorescent compound useful for detecting ROS and determining total oxidative stress levels. We used a fluorescence microscope to observe the staining solution, which was produced according to the instructions and incubated at 37 °C for 20 min (Olympus IX73, Tokyo, Japan). Lipid ROS measurement used C11-BODIPY^581/591^ probe. Briefly, mTEC1 cells were seeded in 12-well plate incubation with/without siRNA, FAC, and L-Cit for 24 h. The C11-BODIPYTM ^581/591^ probe was added for 30 min. Following this, there were three washes with 1 x phosphate-buffered saline. The cells were observed using an Olympus fluorescent microscope (Olympus IX73, Japan).

### 2.12. Immunofluorescence Staining

The sections were fixed for 20 min in 4% tissue fixative solution and rinsed three times in PBS. After that, we employed 0.05% Triton to permeate the membrane for 5 min with 10% goat serum, and the sections were blocked for 90 min at room temperature. mTEC1 cells were seeded on coverslips and fixed with cold methanol for 15 min at room temperature and subsequently permeabilized with 0.1% Triton X-100 and blocking buffer 3% BSA for 1 h at room temperature. Then, cells and sections were incubated with primary antibodies diluted in blocking buffer overnight at 4 °C. After three washes with PBS, the cells were incubated with secondary antibodies diluted in blocking solution for 1 h at room temperature. Cells were then washed three times with PBS, and the coverslips were sealed with nail polish. The nucleus was stained for 5 min with 4′,6-diamidino-2-phenylindole (DAPI) and blocked with antifluorescence quencher. Fluorescence images were acquired using a fluorescence microscope (Olympus IX73) and confocal microscope (Zeiss, Jena, Germany). The primary antibodies including p65 (1:500) and CD8^+^ (1:100), and the secondary antibody Goat anti-Rabbit IgG (H + L) Highly Cross-Adsorbed Alexa Fluor Plus 488 (1:500), were purchased from Proteintech (Wuhan, China).

### 2.13. Statistical Analysis of Data

All the analyses were performed using Graphpad Prism 9, and values are expressed as means ± SEMs. The significance of the differences between the two groups was assessed using a t-test, whereas one-way analysis of variance (ANOVA) was conducted for multiple-group comparisons. The difference marked * *p* < 0.05, ** *p* < 0.01, *** *p* < 0.001, and **** *p* < 0.0001, ns means not significant.

## 3. Results

### 3.1. L-Cit Alleviates Mouse Thymus Damage and Inhibits Iron Deposition

As a central immune organ, the thymus plays an essential role in T lymphocyte differentiation and immunomodulation. To investigate the effect of L-cit on the mouse thymus, the morphology, iron deposition, iron content, and LDH activity were explored. In the control and L-cit-treated group, the division between the cortex and medulla was distinct in the thymus. The defined areas between the cortex and medulla became obviously obscured following iron overload treatment, which meant that the compartmentalization in the thymi degenerated. Moreover, an obvious hemorrhage was observed in the iron overload mice. However, L-cit supplementation could reverse the mouse thymus histologic changes caused by iron overload (Figure 1A). Additionally, iron deposition in the thymus and iron level in the serum were explored. Prussian blue staining demonstrated that L-cit treatment reduced the iron deposition caused by iron overload, and the concentration of serum iron ions was decreased in a dose-dependent manner (Figure 1B,C). LDH activity was further investigated, which can be used as a marker for diverse tissue injuries and reflect tissue-specific pathological conditions or cell death [53]. As expected, L-cit could reverse the LDH elevation induced by iron overload, and high does (2 mg/kg) indicated the significant effects (Figure 1D). These results indicated that L-cit supplementation could ameliorate tissue damage and suppress the iron deposition of thymocytes induced by iron overload.

### 3.2. L-Cit Attenuates Oxidative Stress Induced by Iron Overload in Mouse Thymus

As a specialized primary lymphoid organ of the immune system, the thymus contributes to the maturation, differentiation, and function of T-cells. CD8^+^ T lymphocyte numbers underwent further examination. Immunofluorescence assay indicated that the number of CD8^+^ cells in the iron overload group were lower than that in the control group. Importantly, CD8^+^ cells numbers were elevated with the increasing supplementation of L-cit (Figure 2A,B). It is reported that excessive iron can cause Fenton reaction, produce a large amount of reactive oxygen species (ROS), cause oxidative stress, and show damage to the tissues, which disturb the balance of the antioxidant defense system. The oxidative capacity of L-cit against iron overload in the thymus was detected. Results showed that iron overload treatment can lead to the increased MDA content (Figure 2C) and result in decreased GSH-Px activity (Figure 2D) and SOD activity (Figure 2E). However, L-cit supplementation could elevate the antioxidant capacity by inhibiting MDA content and promoting the SOD and GSH-Px activity in the thymus (Figure 2). Taken together, these results indicated that L-cit supplementation could ameliorate the oxidative stress in the mouse thymus induced by iron overload in a dose-dependent manner.

### 3.3. L-Cit Restrains Ferritinophagy Induced by Iron Overload in Mouse Thymus

NCOA4-mediated ferritinophagy is an autophagic phenomenon that specifically involves ferritin to release intracellular free iron contributing to the maintenance of iron homeostasis. In this study, the expression of ferritin, NCOA4, and GPX4 expression were explored. The results indicated that iron overload elevated the expression of ferritin heavy chain (FTH), which is regarded as an important indicator of iron overload (Figure 3A,B). Moreover, Transferrin 1 (TfR1), a ubiquitously expressed membrane protein, can uptake and release metal ions by interaction with transferrin. Iron overload increases the TfR1 expression (Figure 3A,C). In addition, NCOA4-mediated ferritinophagy and autophagy marker microtubule-associated protein 1 light chain 3 II (LC3 II) are activated (Figure 3D,E). Importantly, GPX4 expression was restrained (Figure 3F). These findings demonstrated that excessive iron enhanced the cellular ferritin level and can be transferred via the transferrin receptor, thereby sensitizing cells to ferritinophagy, which might result in ferroptosis. However, L-cit supplementation could inhibit the expression of ferritin and TfR1. Autophagy markers such as LC3 II and NCOA4 were suppressed by L-cit under iron overload. Particularly, GPX4 expression was elevated (Figure 3). GPX4 was considered as a key component of the ferroptosis regulator, which was reported to convert lipid hydroperoxides to lipid alcohols, and this process prevents the iron (Fe^2+^)-dependent formation of toxic lipid ROS. GPX4 inhibition leads to lipid peroxidation and results in the induction of ferroptosis [54]. These results indicated that L-cit supplementation could ameliorate ferritinophagy and ferroptosis, contributing to the maintenance of the balance of iron metabolism.

### 3.4. L-Cit Suppresses NF-κB Signaling Pathway Induced by Iron Overload in Mouse Thymus

Iron accumulation is mainly due to the inhibition of iron mobilization. Impaired iron export is related to inflammation and metabolic derangements [55]. In this study, inflammatory cytokines and the NF-κB signaling pathway were investigated upon iron overload. Compared with the control group, excess iron likely aggravates inflammatory cytokines expression, which leads to elevated TNF-α (Figure 4A,B), IL-6 (Figure 4A,C), and IL-β (Figure 4A,D) expression upon iron overload. There was no significant effect on p65 expression. However, iron overload robustly upregulated p65 phosphorylation levels, which suggested that the NF-κB signaling pathway was activated in the mouse thymus (Figure 4A,E). We speculated that L-cit could resist the inflammatory response induced by iron overload. Subsequently, we found that L-cit treatment can remarkably inhibit the inflammatory cytokines expression, including TNF-α, IL-6, and IL-β. Importantly, the p65-dependent NF-κB signaling pathway was significantly blocked (Figure 4). These results hinted that L-cit can suppress the NF-κB signaling pathway induced by iron overload, contributing to the attenuation of inflammation in the thymus.

### 3.5. L-Cit Improves Cells’ Viability and Activates Nrf2 Expression in mTEC1 Cells

To investigate the effect of L-cit in vitro against iron overload, the cell viability, iron concentration, and LDH activity were explored in mTEC1 cells. The CCK-8 assay indicated that FAC significantly inhibited the cells’ viability in a dose-dependent manner. FAC was used to construct the cell model of iron overload; 200 μM FAC exerted the significant inhibitory effect at 24 h and 48 h (Figure 5A,B). We further showed that L-cit administration effectively reversed this inhibition (Figure 5C). A 2 mM L-cit treatment was selected as the protective concentration against 200 μM FAC stimulation in the mTEC1 cells (Figure 5D), which can decrease the iron content and LDH activity in mTEC1cells (Figure 5E,F). In addition, MDA content, SOD, and GSH-Px activity were detected. Consistently, antioxidant capacity was promoted following L-cit treatment (Figure 5G–I). Nrf2 functions as a redox-sensitive transcription factor that maintains redox homeostasis by regulating antioxidant-response element (ARE)-dependent transcription and the expression of antioxidant defense enzymes [56] and is responsible for regulating several genes involved in iron metabolism [57]. Thus, Nrf2 expression in cytoplasmic and nuclear was investigated. The data showed that Nrf2 expression in the cytosolic was higher in FAC treatment (Figure 5J,K). However. L-cit treatment enhanced Nrf2 translocation compared to FAC administration (Figure 5J,L), which might contribution to elevating the antioxidant capacity against iron overload. These results demonstrated that the cell viability improvement and antioxidant capacity elevation might rely on Nrf2 activation caused by L-cit treatment in mTEC1 cells.

### 3.6. L-Cit Suppresses Ferritinophagy to Ameliorate FAC-Induced ROS Accumulation in mTEC1 Cells

ROS levels that exceed the capacity of the cellular antioxidant defense system induce oxidative stress. To investigate whether ROS production could be inhibited by L-cit via ferritinophagy, cytoplasmic ROS and lipid peroxidation (lipid ROS) were explored. Firstly, the effective concentration of siNCOA4 was screened using qPCR and western blot. We found that 50 nM siNCOA4 was observed to exert the obvious inhibitory effect, and the interference efficiency exceeded 50% (Figure 6A,B). Compared with the FAC group, siNCOA4 treatment restrained cytoplasmic Nrf2 expression (Figure 6C,D) and elevated the Nrf2 translocation (Figure 6C,E). Consistently, L-cit and siNCOA4 cotreatment could further increase Nrf2 expression in the nucleus. Nrf2 translocation can activate antioxidant signaling by inhibiting ROS production. We thus determined the production of total ROS by using a DCFH-DA fluorescent probe. The staining analysis results showed that after FAC treatment, the ROS level was significantly increased in mTEC1 cells. In contrast, pretreatment of the cells with L-cit and siNCOA4 considerably ameliorated the FAC-mediated overproduction of cytoplasmic ROS (Figure 6F,G). We speculated that L-cit might inhibit lipid peroxidation by suppressing ferritinophagy. Particularly, lipid ROS was measured using a C11 BODIPYTM^581/591^ fluorescent probe, which is often used for indexing lipid peroxidation and antioxidant efficacy in model membrane systems and living cells. Lipid ROS is often applied in the quantitation of ferritinophagy and ferroptosis [58,59]. From Figure 7, Liperfluo signal in oxidized significantly increased in FAC-treated cells, indicating that FAC was able to induce lipid peroxidation in mTEC1 cells. Meanwhile, a reduction in lipid peroxidation was observed in cells pretreated with L-cit and siNCOA4, compared with FAC alone (Figure 7A,B). These results indicated that L-cit ameliorated FAC-induced ROS accumulation in mTEC1 cells by suppressing ferritinophagy.

### 3.7. L-Cit Restrains Ferritinophagy to Prevente Breakdown of Mitochondrial Membrane Potential (MMP) in mTEC1 Cells

MMP plays a key role in vital mitochondrial functions, and its dissipation is a hallmark of mitochondrial dysfunction. To investigate the effect of ferritinophagy regulated by L-cit on MMP in mTEC1 cells induced by FAC, MMP was measured by a JC-1 fluorescent probe. As shown in Figure 7C,D, compared with the control group, the JC-1 aggregates’ fluorescence reduced significantly in the FAC group, which indicated that FAC challenge provoked a loss of membrane potential. Conversely, L-cit and siNCOA4 cotreatment protected the cells against mitochondrial damage after exposure to FAC, as evidenced by the enhanced JC-1 aggregates’ fluorescence intensity (Figure 7). Collectively, the results indicated that L-cit prevented FAC-induced breakdown of MMP by restraining ferritinophagy in mTEC1 cells.

### 3.8. L-Cit Restrains Ferritinophagy to Suppress Ferroptosis Induced by FAC in mTEC1 Cells

NCOA4-mediated ferritinophagy is an autophagic phenomenon that specifically involves ferritin releasing intracellular free iron, which is critical to inducing ferroptosis [60]. To explore the effect of L-cit on ferritinophagy and ferroptosis in mTEC1 cells, TfR1, NCOA4, FTH, GPX4, and LC3 were investigated following L-cit and siNCOA4 with or without cotreatment upon FAC exposure. Compared with the control group, Exogenous FAC elevated the expression of TfR1 (Figure 8A,B), FTH (Figure 8A,C), NCOA4 (Figure 8A,D), and LC3 II (Figure 8A,E) protein were also increased. However, L-cit administration can reverse the effect induced by FAC (Figure 8A). Importantly, as an antioxidant defense enzyme, GPX4 was crucial to reduce membrane lipid hydroperoxide and lipoproteins and functioned as a ferroptosis inhibitor [61]. GPX4 expression was restrained in the FAC-treated group (Figure 8A,F). The subsequent results showed that L-cit and siNCOA4 cotreatment can further rescue GPX4 expression (Figure 8A). Taken together, L-cit can restrain ferritinophagy-mediated ferroptosis induced by FAC in mTEC1 cells.

### 3.9. L-Cit Restrains Ferritinophagy-Mediated Ferroptosis to Alleviate Iron Overload-Induced Inflammation in mTEC1 Cells

It was reported that FAC promoted the inflammation response in immune cells [62] and increased iron stores, which are related to several diseases via impacting inflammation and ROS production [62,63]. We thus detected the NF-κB signaling pathway and the expression of inflammatory factors such as TNFα (Figure 9A,B), IL-6 (Figure 9A,C), and IL-1β (Figure 9A,D) in the mTEC1 cells. We found that FAC upregulated inflammatory factors expression and elevated p65 phosphorylation (Figure 9). L-cit or siNCOA4 treatment alone can reverse the elevation of inflammatory factors expression and p65 phosphorylation (Figure 9A,E). Meanwhile, L-cit and siNCOA4 cotreatment can further inhibit the NF-κB signaling pathway (Figure 9). In addition, nuclear translocation of p65 was observed using an immunofluorescence assay. Endogenous p65 protein was distributed diffusely in both the cytoplasm and cell nuclei in the FAC group; in contrast, L-cit or siNCOA4 treatment alone resulted in the accumulation of p65 around the nucleus and presented a characteristic packed tightly to the nuclear area (Figure 9F). L-cit and siNCOA4 cotreatment exerted more obvious inhibition on nuclear p65 translocation. Collectively, L-cit could restrain ferritinophagy-mediated ferroptosis to alleviate iron overload-induced inflammation in mTEC1 cells.

## 4. Discussions

The thymus is one of the important central immune organs, which contributes to T lymphocyte differentiation and may be important for the shaping of the immune system and adjustment to specific peripheral needs [64]. The thymus is exquisitely sensitive to acute insults such as infection, shock, or common cancer therapies such as cytoreductive chemo- or radiation therapy [65], which results in the damaging of the structure and malfunction, leading to immune dysfunction. As a critical and essential trace element of cellular activity, iron plays a pivotal role in the fight for survival between mammalian hosts and their pathogens [66,67]. Iron homeostasis is essential for health. However, iron overload disorder caused by iron accumulation usually led to oxidative stress and organ damage [68,69]. Importantly, iron overload had the greatest effect on immunity [67]. However, the function of iron overload on the thymus needs to be further investigated. In this study, we found that iron overload resulted in thymus injury, iron excessive deposition, and a drop of antioxidative capacity in vivo and in vitro. The number of CD8^+^ T cells has been further explored in the thymus. CD8^+^ T cells are critical players in cell-mediated adaptive immunity, protecting against microbial infections and inflammation. CD8^+^ T cell-mediated cytotoxicity and tissue injury play an important role in viral, autoimmune, or immune-mediated diseases [70]. Our data demonstrated that the number of CD8^+^ T cells in the thymus was reduced. These results demonstrated that due to iron exposure, iron excessive deposition may aggravate oxidative stress and lead to the decrease in CD8^+^ T cells in the mouse thymus.

As an L-arginine nitric oxide (NO) precursor, L-cit is commonly found in watermelon and has been extensively investigated for its possession of anti-inflammatory and antioxidant properties [71]. Studies indicated that L-cit could improve NO signaling to ameliorate pulmonary hypertension in newborn animals [72,73] and protects gastric mucosa from ischemia–reperfusion injury by scavenging or inhibiting oxygen free radicals [74]. L-cit supplementation could exert anti-inflammation in type 2 diabetes [75] and ulcerative colitis [76]. Moreover, L-cit may improve lung injury by inhibiting NLRP3 inflammasome activation and reducing the release of inflammatory factors [77]. Our results demonstrated that L-cit supplementation has been shown to greatly alleviate thymus hemorrhage and morphological injury and elevate the number of CD8^+^ cells in the mouse thymus in a dose-dependent manner. In addition, L-cit treatment restrains iron deposition in the mouse serum and thymus, which might contribute to its overall action. This function suggested that L-cit might act as an iron chelator to remove iron from the blood or thymus tissue. The results further demonstrated that L-cit could increase antioxidative capacity in the mouse thymus and mTEC1 cells, suggesting that L-cit can hold an important antioxidant role. Importantly, mitochondrial membrane potential was elevated, and cytoplasm ROS and lipid ROS were suppressed following L-cit administration. These findings indicated that L-cit functioned as a protector, exerting the chelator and antioxidant action against iron overload in vivo and in vitro.

Iron is an essential element for all living organisms, and the disturbance of iron homeostasis is associated with altered immune function. Iron is functioned as a powerful regulator of immune responses [78]. Iron overload enhances the immune inflammatory response and accelerates the disease progression. Iron overload is increasingly implicated as a contributor to the pathogenesis of COVID-19 and bacterial infection [79,80]. Inflammation, hypercoagulation, hyperferritinemia, and immune dysfunction are also reminiscent of iron overload [80]. In this study, iron overload resulted in the elevation of inflammatory cytokines expression including TNF-α, IL-6, and IL-β and p65 phosphorylation. However, L-cit administration reversed the inflammatory response and restrained the NF-κB pathway against iron overload. These results indicated that maintaining iron homeostasis is generally beneficial for the immune response.

Ferritinophagy is involved in the regulation of iron homeostasis and is the process of autophagic degradation of the iron-storage protein ferritin, which is critical for the regulation of cellular iron levels [81]. NCOA4-dependent ferritinophagy promotes ferroptosis by releasing free iron from ferritin [82]. Iron-catalyzed lipid peroxidation has been increasingly implicated in the newly discovered non-apoptotic cell death known as ferroptosis [35,80]. Ferroptosis is regarded as an iron-dependent form of oxidative cell death. NCOA4-mediated ferritinophagy is required to maintain intracellular and systemic iron homeostasis [81]. Triggering or blocking autophagy-dependent ferroptosis may be developed for therapeutic interventions in diseases. Autophagy-dependent ferroptosis has been related to tumor growth and macrophage polarization [83]. Activation of CD8^+^ T cells has an ability to enhance the sensitivity of surrounding non-T cells to ferroptosis in cancer therapy [84]. In this study, iron overload resulted in iron accumulation and iron metabolism disorders and induced NCOA4-mediated ferritinophagy and ferroptosis. L-cit administration could reduce the expression of ferritin, TfR1, and LC3, and ferritinophagy was restrained. However, GPX4, a resistance mediator to ferroptosis, saw its expression increased. Particularly, the pharmacological inhibition of ferritinophagy by siRNA significantly mitigated cell ROS levels and inflammation, increasing antioxidative capacity and reserving mitochondrial membrane potential. Moreover, L-cit cooperated with si-NCOA4 and thus exerted a good synergistic effect including anti-inflammation, antioxidant, and ferroptosis inhibition against iron overload.

## 5. Conclusions

In summary, we demonstrated the protective mechanism of L-cit by restraining and ferritinophagy and ferroptosis against iron overload in the mouse thymus. We concluded that L-cit could function as an iron chelator and exert anti-oxidation and anti-inflammation effects by targeting ferritinophagy-mediated ferroptosis, which might be a therapeutic strategy against iron overload-induced thymus oxidative damage and immune dysfunction (Figure 10).

## Figures and Tables

**Figure 1 nutrients-14-04549-f001:**
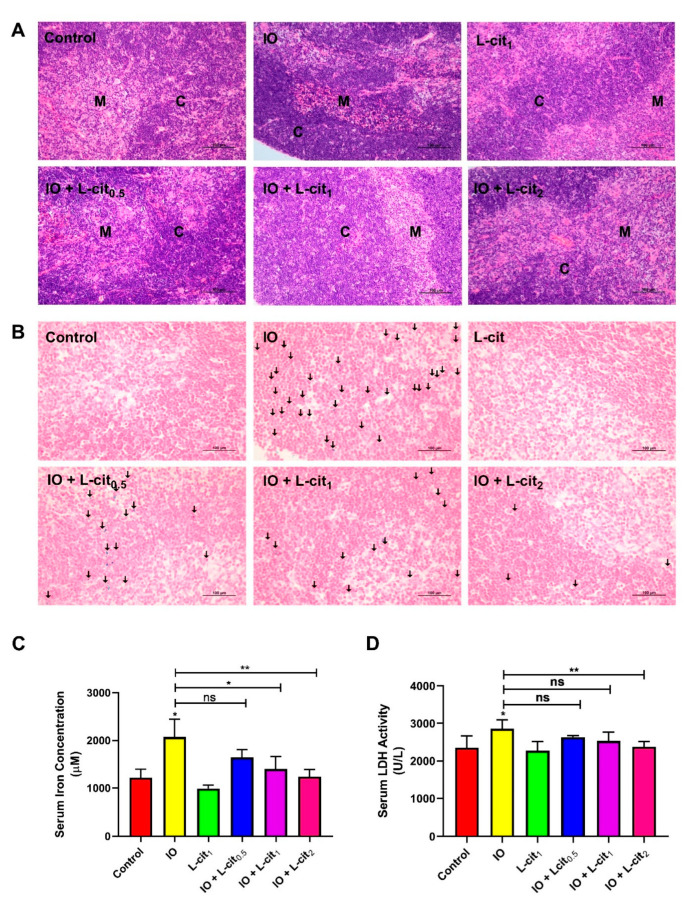
L-cit alleviates mouse thymus damage and inhibits the iron deposition and LDH activity. Mice were intraperitoneally injected with iron dextran 50 mg/kg, and L-cit was given by gavage administration with a dosage of 0.5, 1.0, and 2.0 g/kg/day. (**A**) HE staining of thymus (C represents the thymic cortex, M represents thymus medulla). (**B**) Prussian blue staining of thymus (the arrow indicates iron deposition). (**C**) Iron concentration in serum. (**D**) LDH activity in serum. Values represent mean ± SEM. of three independent experiments. Significant differences are represented by * *p* < 0.05, ** *p* < 0.01, ns (not significant).

**Figure 2 nutrients-14-04549-f002:**
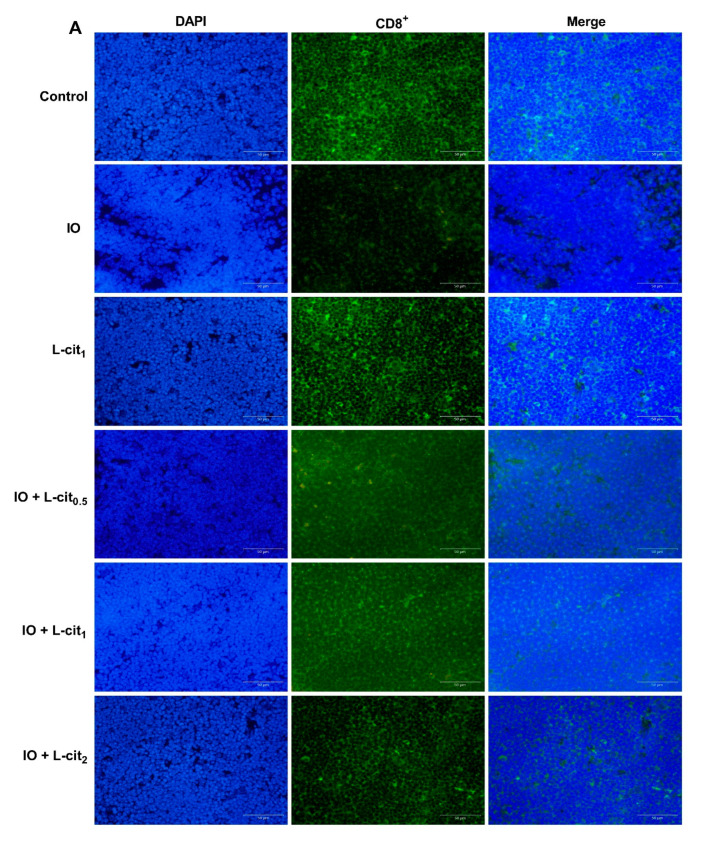
L-cit attenuates oxidative stress induced by iron overload in mouse thymus. Mice were intraperitoneally injected with iron dextran 50 mg/kg, and L-cit was given by gavage administration with a dosage of 0.5, 1.0, and 2.0 g/kg/day. (**A**,**B**) CD8^+^ T-lymphocytes determined by immunofluorescence assay. (**C**) MDA concentration. (**D**) GSH-Px activity. (**E**) SOD activity. Values represent mean ± SEM. of three independent experiments. Significant differences are represented by * *p* < 0.05, ** *p* < 0.01, *** *p* < 0.001, **** *p* < 0.0001, ns (not significant).

**Figure 3 nutrients-14-04549-f003:**
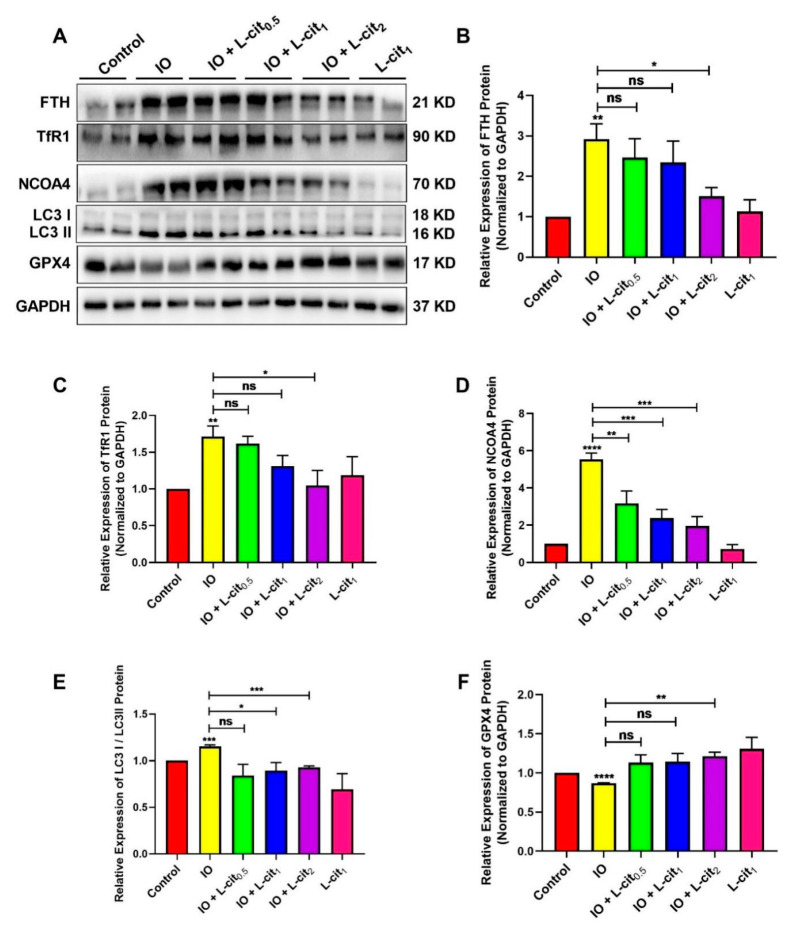
L-cit restrains ferritinophagy induced by iron overload in mouse thymus. Mice were intraperitoneally injected with iron dextran 50 mg/kg, and L-cit was given by gavage administration with a dosage of 0.5, 1.0, and 2.0 g/kg/day. Immunoblot analyses were explored. (**A**,**B**) Immunoblot analyses of FTH. (**A**,**C**) Immunoblot analyses of TfR1. (**A**,**D**) Immunoblot analyses of NCOA4. (**A**,**E**) Immunoblot analyses of LC3. (**A**,**F**) Immunoblot analyses of GPX4. GAPDH was used as the loading control. Values represent mean ± SEM. of three independent experiments. Significant differences are represented by * *p* < 0.05, ** *p* < 0.01, *** *p* < 0.001, **** *p* < 0.0001, ns (not significant).

**Figure 4 nutrients-14-04549-f004:**
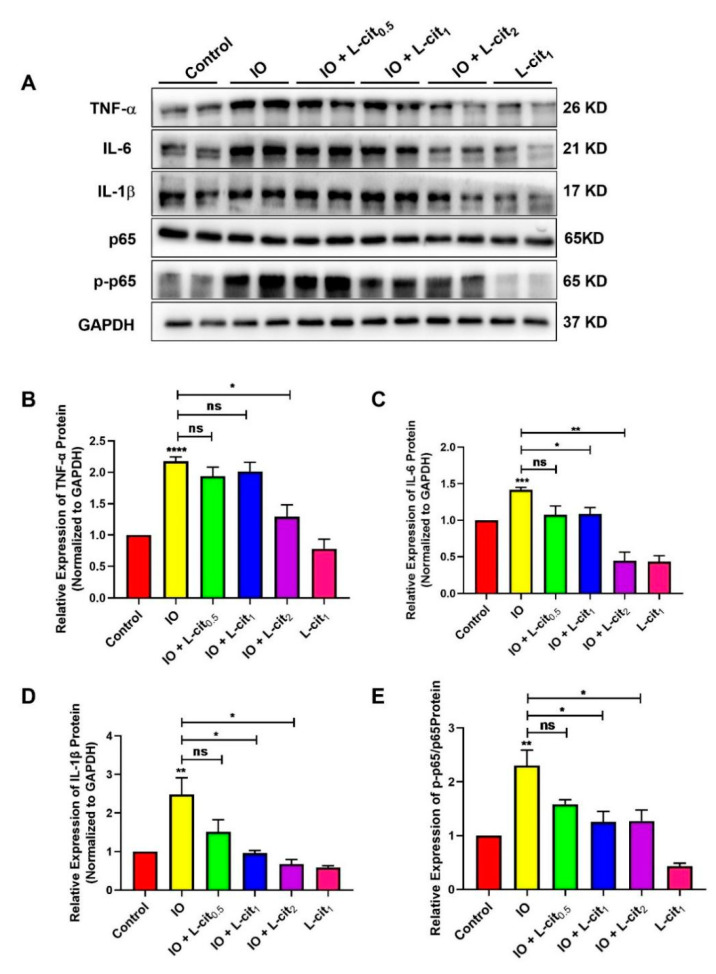
L-cit suppresses NF-κB signaling pathway induced by iron overload in mouse thymus. Mice were intraperitoneally injected with iron dextran 50 mg/kg, and L-cit were given by gavage administration the dosage of 0.5, 1.0, and 2.0 g/kg/day. Immunoblot analyses were explored. (**A**,**B**) Immunoblot analyses of TNF-α. (**A**,**C**) Immunoblot analyses of IL-6. (**A**,**D**) Immunoblot analyses of IL-1β. (**A**,**E**) Immunoblot analyses of p-p65/p65. GAPDH was used as the loading control. Values represent mean ± SEM. of three independent experiments. Significant differences are represented by * *p* < 0.05, ** *p* < 0.01, *** *p* < 0.001, **** *p* < 0.0001, ns (not significant).

**Figure 5 nutrients-14-04549-f005:**
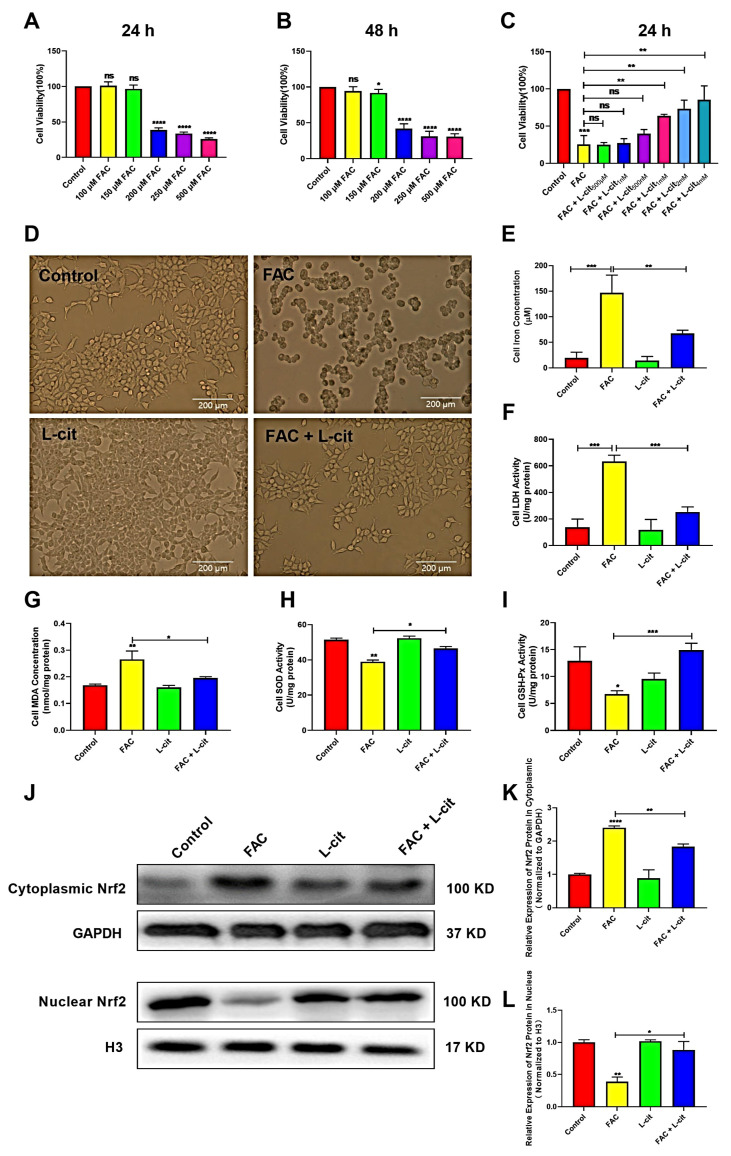
L-cit improves the cell viability, antioxidant capacity, and Nrf2 activation in the mTEC1 cells. The mTEC1cells were exposed to 100, 150, 200, 250, and 500 μM FAC for 24 h or 48 h. Cell viability, iron concentration, LDH activity, antioxidant capacity, and Nrf2 expression were investigated. (**A**) Cell viability at 24 h following FAC treatment. (**B**) Cell viability at 48 h following FAC treatment. (**C**) Cell viability at 24 h following 200 μM FAC cotreated with different dosage of L-cit. (**D**) Cell morphology and viability treated with 2 mM L-cit, followed by 200 μM FAC for 24 h. (**E**) Iron concentration. (**F**) LDH activity. (**G**) MDA concentration. (**H**) SOD activity. (**I**) GSH-Px activity. (**J**,**K**) Immunoblot analyses of Nrf2 in cytoplasm. GAPDH was used as the loading control. (**J**,**L**) Immunoblot analyses of Nrf2 in nucleus. H3 (Histone H3) was used as the loading control. Values represent mean ± SEM. of three independent experiments. Significant differences are represented by * *p* < 0.05, ** *p* < 0.01, *** *p* < 0.001, **** *p* < 0.0001, ns (not significant).

**Figure 6 nutrients-14-04549-f006:**
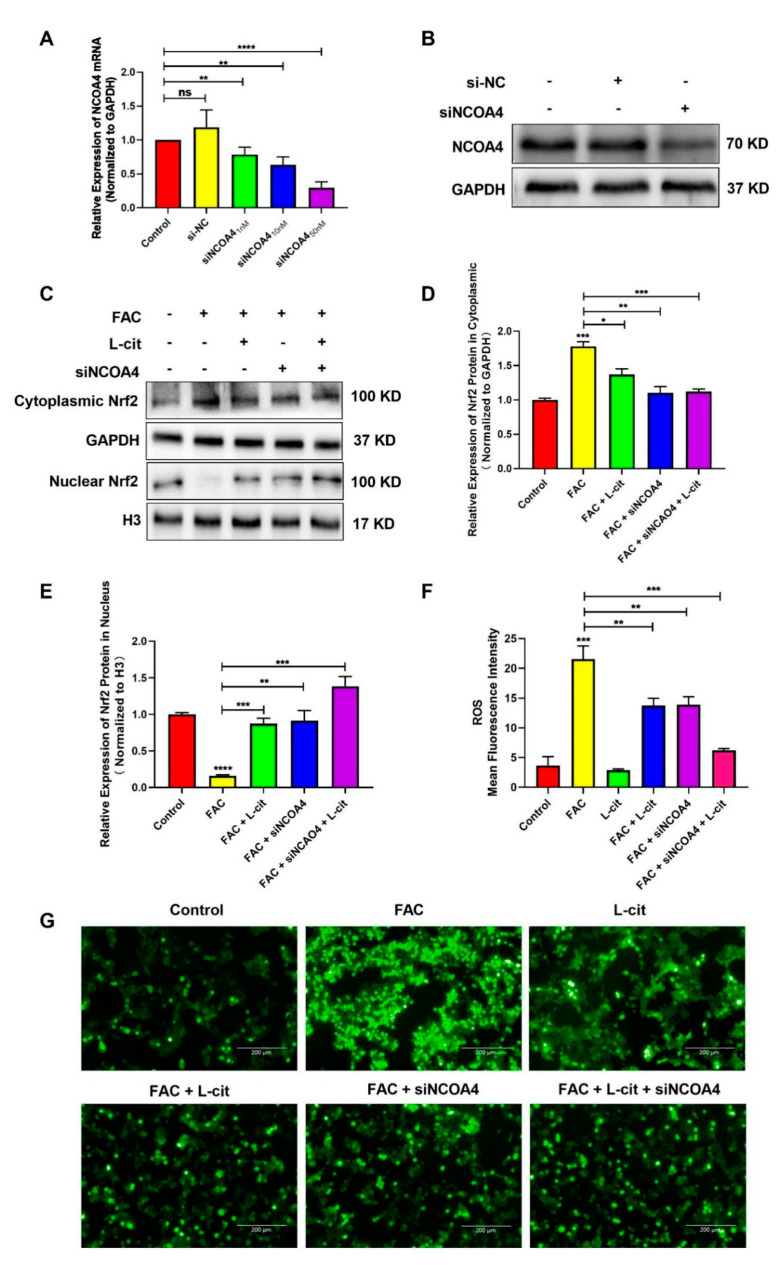
L-cit suppresses ferritinophagy to ameliorate FAC-induced cytoplasmic ROS accumulation in the mTEC1 cells. (**A**) Determination of siNCOA4 concentration. mTEC1 cells were transfected with negative control siRNA or different concentration of siNCOA4 to detect the NCOA4 mRNA level. (**B**) Confirmation of siNCOA4 concentration by qPCR and NCOA4 protein quantitation by western blotting after transfection with 50 nM siNCOA4. The mTEC1 cells were transfected with 50 nM siNCOA4, followed by 2 mM L-cit, and treated with 200 μM FAC for 24 h. (**C**,**D**) Immunoblot analyses of Nrf2 in cytoplasm. GAPDH were used as the loading control. (**C**,**E**) Immunoblot analyses of Nrf2 in nucleus. H3 was used as the loading control. (**F**) ROS mean fluorescence intensity. (**G**) Cytoplasmic ROS level. Values represent mean ± SEM. of three independent experiments. Significant differences are represented by * *p* < 0.05, ** *p* < 0.01, *** *p* < 0.001, **** *p* < 0.0001, ns (not significant).

**Figure 7 nutrients-14-04549-f007:**
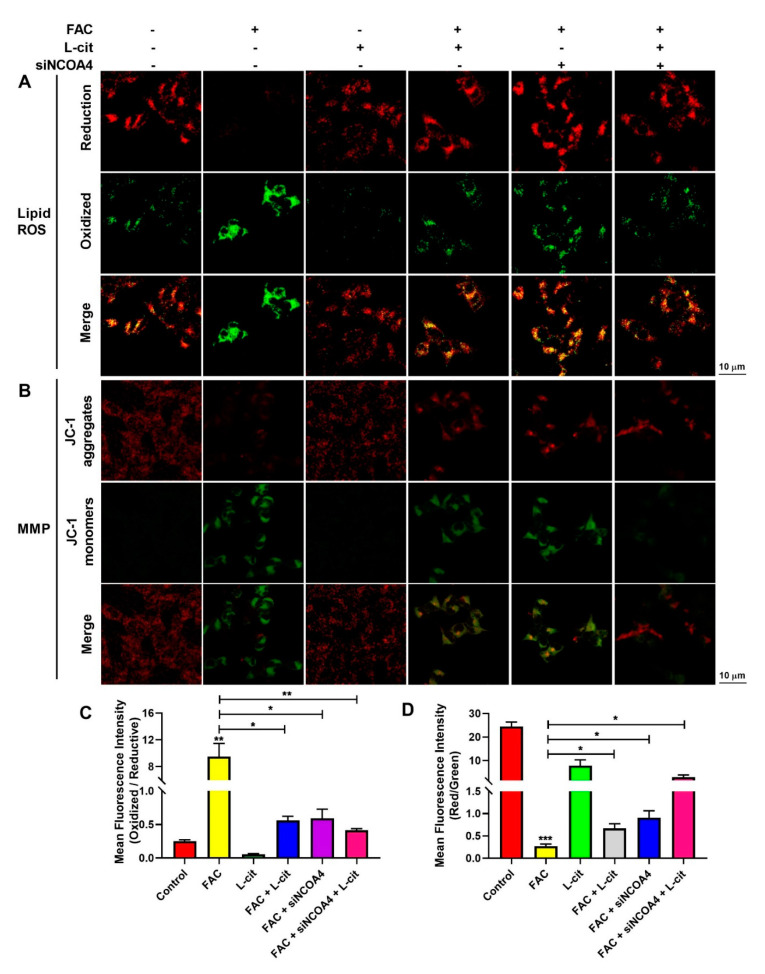
L-cit suppresses ferritinophagy to ameliorate FAC-induced lipid ROS accumulation and MMP elevation in mTEC1 cells. The mTEC1 cells were transfected with 50 nM siNCOA4, followed by 2 mM L-cit, and treated with 200 μM FAC for 24 h. (**A**,**B**) Lipid ROS detection using C11 BODIPYTM^581/591^ fluorescent probe. Green fluorescence indicates oxidized and red fluorescence indicates reduction. (**C**,**D**) Mitochondrial membrane potential (MMP) detection by JC-1 fluorescent probe. Green fluorescence indicates JC-1 monomer and red fluorescence indicates JC-1 aggregate. Values represent mean ± SEM. of three independent experiments. Significant differences are represented by * *p* < 0.05, ** *p* < 0.01, *** *p* < 0.001.

**Figure 8 nutrients-14-04549-f008:**
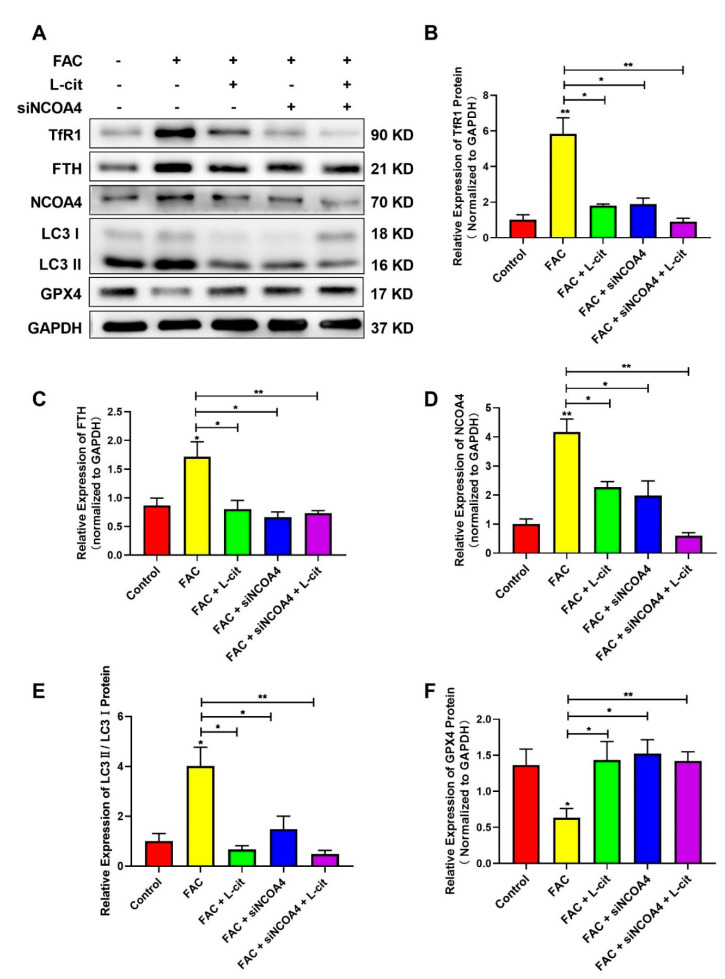
L-cit restrains ferritinophagy to suppress ferroptosis induced by FAC in mTEC1 cells. The mTEC1 cells were transfected with 50 nM siNCOA4, followed by 2 mM L-cit, and treated with 200 μM FAC for 24 h. (**A**,**B**) Immunoblot analyses of TFR1. (**A**,**C**) Immunoblot analyses of FTH. (**A**,**D**) Immunoblot analyses of NCOA4. (**A**,**E**) Immunoblot analyses of LC3. (**A**,**F**) Immunoblot analyses of GPX4. GAPDH was used as the loading control. Values represent mean ± SEM. of three independent experiments. Significant differences are represented by * *p* < 0.05, ** *p* < 0.01.

**Figure 9 nutrients-14-04549-f009:**
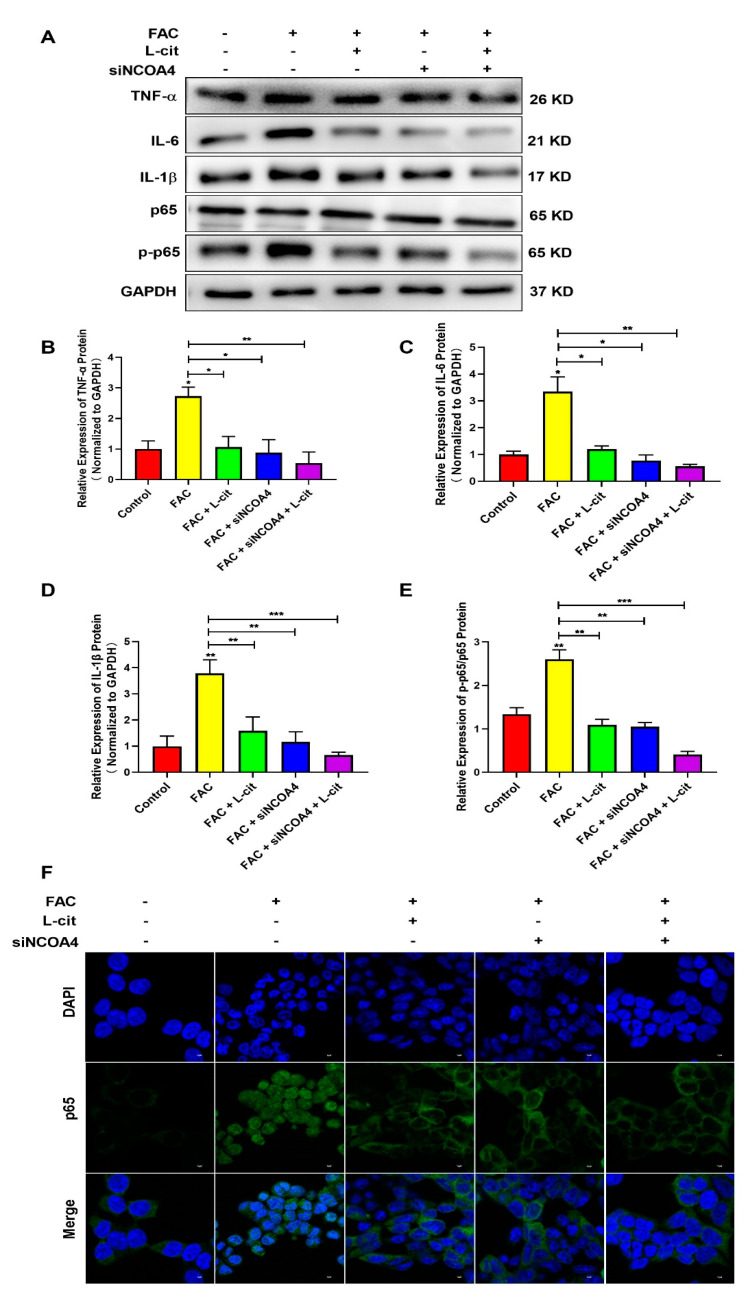
L-cit restrains ferritinophagy-mediated ferroptosis to alleviate inflammation in mTEC1 cells. The mTEC1 cells were transfected with 50 nM siNCOA4, followed by 2 mM L-cit, and treated with 200 μM FAC for 24 h. Immunoblot analyses were explored. (**A**,**B**) Immunoblot analyses of TNF-α. (**A**,**C**) Immunoblot analyses of IL-6. (**A**,**D**) Immunoblot analyses of IL-1β. (**A**,**E**) Immunoblot analyses of p-p65/p65. (**F**) Detection of p65 nuclear translocation using immunofluorescence assay. GAPDH was used as the loading control. Values represent mean ± SEM. of three independent experiments. Significant differences are represented by * *p* < 0.05, ** *p* < 0.01, *** *p* < 0.001.

**Figure 10 nutrients-14-04549-f010:**
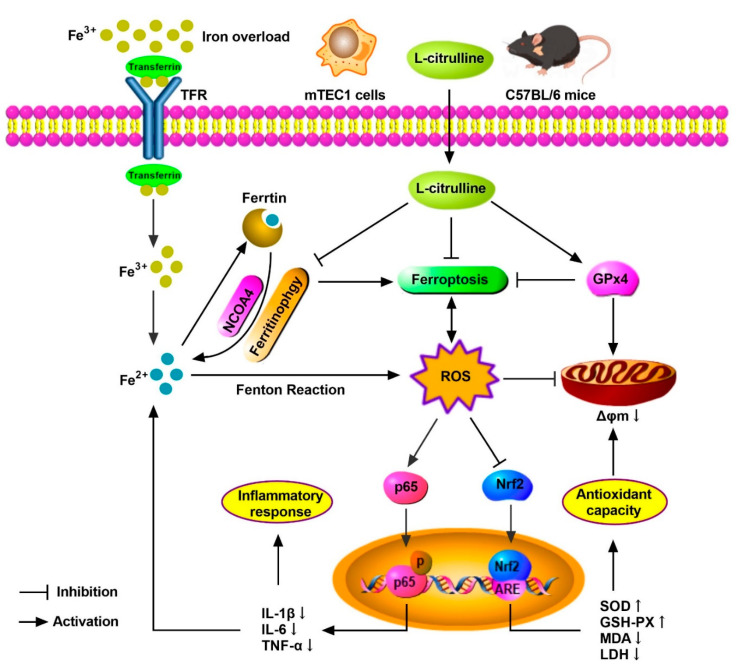
The schematic illustration of underlying mechanism of L-cit alleviating oxidative damage and immune dysfunction. As a result, L-cit alleviates thymus histological damage, reduces the iron deposition, improves antioxidative capacity, and restrains NF-κB pathway in the mouse thymus and mTEC1 cells. NCOA4-mediated ferritinophagy and ferroptosis were attenuated. L-cit might target ferritinophagy-mediated ferroptosis to decrease iron deposition and exert antioxidation and anti-inflammation response, which could be a therapeutic strategy against iron overload-induced oxidative damage and immune dysfunction.

## Data Availability

The data presented in this study are available on request from the corresponding authors.

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
