# Peer review of "L-Citrulline Supplementation Restrains Ferritinophagy-Mediated Ferroptosis to Alleviate Iron Overload-Induced Thymus Oxidative Damage and Immune Dysfunction"

_nutrients, 2022, doi:10.3390/nu14214549_

Round 1
Reviewer 1 Report
Dear Editor, Dear Authors,
I was invited to evaluate the manuscript « L-citrulline supplementation restrains ferritinophagy-mediated ferroptosis to alleviate iron overload-induced thymus oxidative damage and immune dysfunction » by Tongtong Ba et al.
In this study, the authors evaluated the effect of citrulline supplementation in ferroptosis in relation to oxidative stress and immune alterations. Although effects of L-citrulline (L-cit) as antioxidant and antiinflammation are known, its effects on iron overload in the thymus remains unclear. To clarify this point, authors evaluated the antioxidant and antiinflammation effect of L-cit on iron overload-induced in the thymus in animals. Data demonstrate that L-cit administration could inhibit thymus histological damage with a parallel reducing in iron deposition. Authors furthermore demonstrate that various inflammatory pathways were inhibited too : NF-κB pathway, NCOA4 mediated-ferritinophagy, and ferroptosis. In addition, using NCOA4 knockdown, authors demonstrated that this knockdown could reduce the intracellular cytoplasmic ROS, in relation to the Nfr2 activation, reducing although ferroptosis and lipid ROS, with a parallel increase in mitochondrial membrane potential. Finally, the inhibition of NF-κB pathway by L-cit limited ferritinophagy-mediated ferroptosis. Conclusions of the authors were that L-cit inhibits ferritinophagy-mediated ferroptosis through antioxidant and antiinflammation effects making it an option strategy against iron overload-induced thymus damages and immune alterations.
Major comments :
1- The authors must explain why focusing on the thymus in their study on iron damages. In fact, the authors them-self stated in the introduction (Lines 58-59) : « . Iron deposition in most organs has been described, the study in the thymus is rarely observed. The mechanism of thymic immune damage induced by iron overload is unclear. ». If iron damage of the thymus is rarely described, well why focusing on it. Please add the references showing that iron overload is able indeed of causing thymus injuries, if not, the study is not relevant. Does any clinical observation supports thymus injuries during iron overload ?
2- Authors showed that L-cit not only protected thymus but also decrease iron content in serum. This needs to be further commented. Indeed, the fact that L-cit decreases serum concentration of Iron suggests in act also either on its absorption (not relevant here as iron was given through ip injection) and/or elimination or that L-cit may chelate iron removing it from the blood. The fact that Lcit decreases iron in the serum (and body) is enough to inhibit its effects on the body and thymus (an iron chelator would also protect the tissues without the need of antioxidant/antiinflammatory action). Please comment, also in Fig 10.
3- Figure 5 : in vivo assays were performed with L-cit expressed as g/kg. In Figure 5 and related tests, L-cit is expressed in mM. Please indicate in the text to which massic concentration corresponds 2 mM. Although I understand that it is not possible to have the exact volume of mice, please make an estimation so to be able to compare the 2 mM used to the 0.5-2 g/kg used in vivo.
Minor comments :
Line 252 : « and high does (2mg/kg) » please correct
Figure 4 : figure is cut at the level of Fig 4D/E
regards
Author Response
Reviewer: 1
We would like to thank the respected reviewer 1 for his useful comments. Those comments are all valuable and very helpful for revising and improving our paper, as well as the important guiding significance to our researches. We have studied comments carefully and have made correction which we hope meet with approval.
Major comments:
1-The authors must explain why focusing on the thymus in their study on iron damages. In fact, the authors them-self stated in the introduction (Lines 58-59) : « . Iron deposition in most organs has been described, the study in the thymus is rarely observed. The mechanism of thymic immune damage induced by iron overload is unclear. ». If iron damage of the thymus is rarely described, well why focusing on it. Please add the references showing that iron overload is able indeed of causing thymus injuries, if not, the study is not relevant. Does any clinical observation supports thymus injuries during iron overload?
Response
We thank the reviewer for pointing this issue. The relevant references have been added in the introduction.
It is demonstrated that iron is a vital metal for the proliferation of all cells including those of the immune system. Iron deficiency contributes to a reduction in peripheral T cells and atrophy of the thymus, and iron deficiency on the thymus is well established [1]. However, the relationship between iron overload and damage of the thymus is rarely described. Iron overload is one of the factors that cause oxidative stress and immune dysfunction. Antioxidant activity elevation in the thymus improves thymus atrophy [2], Moreover, increases in circulating CD4+ T cells and CD4/CD8 ratios, as well as impaired T cell responses to mitogens, have all been linked to transfusional iron excess in -thalassemia [3]. Circulating CD1a+ T cells point to a thymic dysfunction that may be caused by the iron excess[1, 4]. However, the mechanism of thymic immune damage induced by iron overload need to be further investigated.
2- Authors showed that L-cit not only protected thymus but also decrease iron content in serum. This needs to be further commented. Indeed, the fact that L-cit decreases serum concentration of Iron suggests in act also either on its absorption (not relevant here as iron was given through ip injection) and/or elimination or that L-cit may chelate iron removing it from the blood. The fact that Lcit decreases iron in the serum (and body) is enough to inhibit its effects on the body and thymus (an iron chelator would also protect the tissues without the need of antioxidant/antiinflammatory action). Please comment, also in Fig 10.
Response
We thank the reviewer for pointing this issue.
Trivalent iron are not bioavailableand and need to be reduced to Fe2+ ( divalent iron ions ) under the action of Dcytb ( duodenal enzyme cytochrome b reductase ). The iron was intaken by intestinal cells and is preferentially deposited in ferritin [5]. Iron in the cytoplasm mainly exists in the form of ferrous iron, or binds to GSH and is transported by bingding to transferrin (Tf) [6, 7]. Under the action of transferrin receptor, iron is detected and then transported to other tissues [8]. In contrast, extracellular iron has been converted to the Fe2+ state by STEAP family reductases and enters cells directly by substituting for surface transporters [9]. In the current study, L-cit may chelate iron removing it from the blood and inhibit its effects on the body and thymus. Although an iron chelator would also protect the tissues, L-cit exerts antioxidant/antiinflammatory action in the thymus from the results.
3- Figure 5 : in vivo assays were performed with L-cit expressed as g/kg. In Figure 5 and related tests, L-cit is expressed in mM. Please indicate in the text to which massic concentration corresponds 2 mM. Although I understand that it is not possible to have the exact volume of mice, please make an estimation so to be able to compare the 2 mM used to the 0.5-2 g/kg used in vivo.
Response
We thank the reviewer for pointing this issue.
In vivo assays, L-cit were given by gavage administration every day according to the mice weight and L-cit was expressed as g/kg. In vitro assay (Figure 5 and related tests), L-cit was added into cell culture medium and is expressed in mM. These are in vivo and in vitro assays.
Minor comments:
1-Line 252: « and high does (2 mg/kg) » please correct
Response
We thank the reviewer for pointing this issue.
We have corrected in the revised manuscript.
2-Figure 4: figure is cut at the level of Fig 4D/E
Response
We thank the reviewer for pointing this issue.
The full image was supplied in the revised manuscript.
References
- Kuvibidila S, Dardenne M, Savino W, Lepault F: Influence of iron-deficiency anemia on selected thymus functions in mice: thymulin biological activity, T-cell subsets, and thymocyte proliferation. The American Journal of Clinical Nutrition 1990, 51:228-232.
- Griffith AV, Venables T, Shi J, Farr A, van Remmen H, Szweda L, Fallahi M, Rabinovitch P, Petrie HT: Metabolic Damage and Premature Thymus Aging Caused by Stromal Catalase Deficiency. Cell Rep 2015, 12:1071-1079.
- Guglielmo P, Cunsolo F, Lombardo T, Sortino G, Giustolisi R, Cacciola E, Cacciola E: T-Subset Abnormalities in Thalassaemia intermedia: Possible Evidence for a Thymus Functional Deficiency. Acta Haematologica 1984, 72:361-367.
- Guglielmo P, Cunsolo F, Lombardo T, Sortino G, Giustolisi R, Cacciola E, Cacciola E: T-subset abnormalities in thalassaemia intermedia: possible evidence for a thymus functional deficiency. Acta Haematol 1984, 72:361-367.
- Theil EC: Ferritin: structure, gene regulation, and cellular function in animals, plants, and microorganisms. Annu Rev Biochem 1987, 56:289-315.
- Hentze MW, Muckenthaler MU, Andrews NC: Balancing acts: molecular control of mammalian iron metabolism. Cell 2004, 117:285-297.
- Philpott CC, Patel SJ, Protchenko O: Management versus miscues in the cytosolic labile iron pool: The varied functions of iron chaperones. Biochim Biophys Acta Mol Cell Res 2020, 1867:118830.
- Badu-Boateng C, Naftalin RJ: Ascorbate and ferritin interactions: Consequences for iron release in vitro and in vivo and implications for inflammation. Free Radic Biol Med 2019, 133:75-87.
- Lane DJ, Merlot AM, Huang ML, Bae DH, Jansson PJ, Sahni S, Kalinowski DS, Richardson DR: Cellular iron uptake, trafficking and metabolism: Key molecules and mechanisms and their roles in disease. Biochim Biophys Acta 2015, 1853:1130-1144.

Reviewer 2 Report
In their manuscript, the authors studied the antioxidative and anti-inflammatory effects of L-citrulline supplementation on iron overload induced in the thymus. The authors showed that L-cit can prevent thymus damage and reduce iron deposits. Their results suggest that the mechanism involves NCOA4 which regulates ferrinitophagy and is required for ferroptosis. They conclude that L-cit might target ferritinophagy and ferroptosis to exert antioxidant and anti-inflammatory functions.
The study is well conducted and the experiments are well designed. However, the following comments need to be addressed for this manuscript to be suitable for publication:
Major comments:
- English should be improved throughout the text.
-Fig1A: the definition of the letters “C” and “M” is not given in the figure legend and what the arrows indicate is not explained either. The legend needs to contain more information about what has been done and what each panel shows.
-Fig2A: the quality of the images is not optimal and could be improved, especially for the merged images in which the DAPI staining is way too strong to clearly see the CD8 staining.
-Fig3: for non-expert readers, the authors should explain briefly in the result section the role of GPX4.
- Result section for Fig5: please explain what FAC is and why you are looking at it.
- Result section for Fig7: please explain what MMP is and why you are looking at it.
Minor comments:
Line 35-38: the sentence seems to be incomplete and needs to be corrected.
Fig4 overlaps with the text and is truncated at the bottom.
Fig5C: on the x-axis, the legend of the bars seems to be disorganized with L-cit concentrations (500 uM on the left of 1nM and then followed on the right by 500 nM, 1mM and 4mM). Please correct the organization if needed.
Author Response
Reviewer: 2
We would like to thank the respected reviewer 1 for his useful comments. Those comments are all valuable and very helpful for revising and improving our paper, as well as the important guiding significance to our researches. We have studied comments carefully and have made correction which we hope meet with approval.
Major comments:
1- English should be improved throughout the text.
Response
We thank the reviewer for pointing this issue.
We have modified language throughout the text as appropriate. The revisions are marked in red.
2-Fig1A: the definition of the letters “C” and “M” is not given in the figure legend and what the arrows indicate is not explained either. The legend needs to contain more information about what has been done and what each panel shows.
Response
We thank the reviewer for pointing this issue.
In the Fig 1A, C represents the thymic cortex, M represents thymus medulla. The more information has been added tin the legend with marked in red.
3-Fig2A: the quality of the images is not optimal and could be improved, especially for the merged images in which the DAPI staining is way too strong to clearly see the CD8 staining.
Response
We thank the reviewer for pointing this issue.
In the current study, immunofluorescence assay was conducted by using thymus tissues. There are many thymus cells in one view and the thymus nucleus is large, which resulted in DAPI staining too deep. We try to improve this problem and the high-quality images were selected in the manuscript.
4-Fig3: for non-expert readers, the authors should explain briefly in the result section the role of GPX4.
Response
We thank the reviewer for pointing this issue.
We have explained the role of GPX4 in the result section in Fig 3 with marked in red.
5- Result section for Fig5: please explain what FAC is and why you are looking at it.
Response
We thank the reviewer for pointing this issue.
FAC is the abbreviation of Ferric Ammonium Citrate. FAC is often used as a food additive for iron supplementation, which can induce ferroptosis [1]. In the current study, FAC was used to construct the cell model of iron overload. We have explained in the manuscript with marked in red.
6- Result section for Fig7: please explain what MMP is and why you are looking at it.
Response
We thank the reviewer for pointing this issue.
MMP is the abbreviation of Mitochondrial Membrane Potential. MMP plays a key role in vital mitochondrial functions, and its dissipation is a hallmark of mitochondrial dysfunction [2]. In this study, Oxidative damage caused by FAC led to a loss of mitochondrial membrane potential in mTEC1 cells. Conversely, L-cit and siNCOA4 cotreatment protected the cells against mitochondrial damage after exposure to FAC, as evidenced by enhanced JC-1 aggregates fluorescence intensity.
Minor comments:
1-Line 35-38: the sentence seems to be incomplete and needs to be corrected.
Response
We thank the reviewer for pointing this issue.
The sentence has been corrected in the revised manuscript with marked in red.
2-Fig4 overlaps with the text and is truncated at the bottom.
Response
We thank the reviewer for pointing this issue.
The full figure was supplied in the revised manuscript.
References
- Wu, W.; Geng, Z.; Bai, H.; Liu, T.; Zhang, B. Ammonium Ferric Citrate induced Ferroptosis in Non-Small-Cell Lung Carcinoma through the inhibition of GPX4-GSS/GSR-GGT axis activity. International journal of medical sciences 2021, 18, 1899-1909, doi:10.7150/ijms.54860
- Audi, S.H.; Cammarata, A.; Clough, A.V.; Dash, R.K.; Jacobs, E.R. Quantification of mitochondrial membrane potential in the isolated rat lung using rhodamine 6G. Journal of applied physiology (Bethesda, Md. : 1985) 2020, 128, 892-906, doi:10.1152/japplphysiol.00789.2019

Round 2
Reviewer 1 Report
Dear Authors, dear Editor,
The raised concern about interpretation and comments to the reduced level of iron in treated animals has not been addressed. Please add a comment to the results or discussion indicating that in addition to antioxidant effect, L-cit also reduce iron level into body, an effect that can contribute to its overall action.
2- Authors showed that L-cit not only protected thymus but also decrease iron content in serum. This needs to be further commented. Indeed, the fact that L-cit decreases serum concentration of Iron suggests in act also either on its absorption (not relevant here as iron was given through ip injection) and/or elimination or that L-cit may chelate iron removing it from the blood. The fact that Lcit decreases iron in the serum (and body) is enough to inhibit its effects on the body and thymus (an iron chelator would also protect the tissues without the need of antioxidant/antiinflammatory action). Please comment, also in Fig 10.
Response
We thank the reviewer for pointing this issue.
Trivalent iron are not bioavailableand and need to be reduced to Fe2+ ( divalent iron ions ) under the action of Dcytb ( duodenal enzyme cytochrome b reductase ). The iron was intaken by intestinal cells and is preferentially deposited in ferritin [5]. Iron in the cytoplasm mainly exists in the form of ferrous iron, or binds to GSH and is transported by bingding to transferrin (Tf) [6, 7]. Under the action of transferrin receptor, iron is detected and then transported to other tissues [8]. In contrast, extracellular iron has been converted to the Fe2+ state by STEAP family reductases and enters cells directly by substituting for surface transporters [9]. In the current study, L-cit may chelate iron removing it from the blood and inhibit its effects on the body and thymus. Although an iron chelator would also protect the tissues, L-cit exerts antioxidant/antiinflammatory action in the thymus from the results.
Author Response
The raised concern about interpretation and comments to the reduced level of iron in treated animals has not been addressed. Please add a comment to the results or discussion indicating that in addition to antioxidant effect, L-cit also reduce iron level into body, an effect that can contribute to its overall action.
Response
We thank the reviewer for pointing this issue. The comments have been added in the discussion in line 1145-1152.
L-cit treatment restrains iron deposition in mice serum and thymus, which might contribute to its overall action in the body. This function suggested that L-cit might act as an iron chelator to remove iron from the blood or tissues. The results further demonstrated that L-cit could increase antioxidative capacity in mice thymus and mTEC1 cells, suggesting that L-cit hold an important role in antioxidant. Importantly, mitochondrial membrane potential was elevated, and cytoplasm ROS and lipid ROS were suppressed following L-cit administration. These findings indicated that L-cit functioned as a protector exerting the iron chelator and antioxidant action against iron overload in vivo and vitro.
The comments have been revised in conclusion and in Fig 10.
Line 1184-1188:
We concluded that L-cit could function as an iron chelator and exert anti-oxidation and anti-inflammation effects by targeting ferritinophagy-mediated ferroptosis, which might be a therapeutic strategy against iron overload-induced oxidative damage and thymus immune dsyfunction (Figure 10).
Line 1248-1251:
L-cit might target ferritinophagy-mediated ferroptosis to decrease iron deposition and exert antioxidation and anti-inflammation response, which could be a therapeutic strategy against iron overload-induced oxidative damage and immune dysfunction.